# Biallelic Variants in *ENDOG* Associated with Mitochondrial Myopathy and Multiple mtDNA Deletions

**DOI:** 10.3390/cells11060974

**Published:** 2022-03-12

**Authors:** Alessia Nasca, Andrea Legati, Megi Meneri, Melisa Emel Ermert, Chiara Frascarelli, Nadia Zanetti, Manuela Garbellini, Giacomo Pietro Comi, Alessia Catania, Costanza Lamperti, Dario Ronchi, Daniele Ghezzi

**Affiliations:** 1Unit of Medical Genetics and Neurogenetics, Fondazione IRCCS Istituto Neurologico Carlo Besta, 20126 Milan, Italy; alessia.nasca@istituto-besta.it (A.N.); andrea.legati@istituto-besta.it (A.L.); melisaemel.ermert@studenti.unimi.it (M.E.E.); chiara.frascarelli@istituto-besta.it (C.F.); nadia.zanetti@istituto-besta.it (N.Z.); alessia.catania@istituto-besta.it (A.C.); costanza.lamperti@istituto-besta.it (C.L.); 2Neurology Unit, Fondazione IRCCS Ca’ Granda Ospedale Maggiore Policlinico, 20122 Milan, Italy; megimeneri@gmail.com (M.M.); manuela.garbellini@policlinico.mi.it (M.G.); dario.ronchi@unimi.it (D.R.); 3Department of Pathophysiology and Transplantation (DEPT), University of Milan, 20122 Milan, Italy; giacomo.comi@unimi.it; 4Neuromuscular and Rare Diseases Unit, Fondazione IRCCS Ca’ Granda Ospedale Maggiore Policlinico, 20122 Milan, Italy

**Keywords:** endonuclease G, ENDOG, mitochondrial DNA, mitochondrial myopathy, multiple mtDNA deletions

## Abstract

Endonuclease G (ENDOG) is a nuclear-encoded mitochondrial-localized nuclease. Although its precise biological function remains unclear, its proximity to mitochondrial DNA (mtDNA) makes it an excellent candidate to participate in mtDNA replication, metabolism and maintenance. Indeed, several roles for ENDOG have been hypothesized, including maturation of RNA primers during mtDNA replication, splicing of polycistronic transcripts and mtDNA repair. To date, *ENDOG* has been deemed as a determinant of cardiac hypertrophy, but no pathogenic variants or genetically defined patients linked to this gene have been described. Here, we report biallelic *ENDOG* variants identified by NGS in a patient with progressive external ophthalmoplegia, mitochondrial myopathy and multiple mtDNA deletions in muscle. The absence of the ENDOG protein in the patient’s muscle and fibroblasts indicates that the identified variants are pathogenic. The presence of multiple mtDNA deletions supports the role of ENDOG in mtDNA maintenance; moreover, the patient’s clinical presentation is very similar to mitochondrial diseases caused by mutations in other genes involved in mtDNA homeostasis. Although the patient’s fibroblasts did not present multiple mtDNA deletions or delay in the replication process, interestingly, we detected an accumulation of low-level heteroplasmy mtDNA point mutations compared with age-matched controls. This may indicate a possible role of ENDOG in mtDNA replication or repair. Our report provides evidence of the association of *ENDOG* variants with mitochondrial myopathy.

## 1. Introduction

Endonuclease G (ENDOG) is a nuclear-encoded nuclease, member of the conserved DNA/RNA non-specific ββα-Me-finger nuclease family [1]. The ENDOG protein is synthesized in the cytoplasm as an inactive 33 kDa propeptide, which is activated by proteolytic cleavage of the mitochondrial targeting sequence, thus producing a mature 28 kDa enzyme which acts as a homodimer [1,2,3]. 

Initial experiments suggested an exclusive localization of ENDOG within the mitochondrial intermembrane space; later on, it was found to be mainly bound to the mitochondrial inner membrane, facing the matrix [4]. Given its spatial closeness to mtDNA, as well as its ability to cleave nucleic acids (DNA, RNA and hybrid molecules) at double-stranded (dG)n·(dC)n and single-stranded (dC)n tracts tested in vitro [1], ENDOG is an excellent candidate to be involved in mtDNA replication initiation, mtRNA processing and mtDNA maintenance and homeostasis [5]. It has been reported to be involved in several biological processes, including apoptosis, paternal mitochondrial elimination in early embryogenesis and autophagy. Previous studies have reported a possible role of ENDOG as one of the apoptotic nucleases that move from the mitochondria to the nucleus, leading to DNA fragmentation [6]. However, none of the *ENDOG* null models (yeast, worm or mouse) provide convincing evidence of significant effects on apoptosis [7,8,9], or in paternal mitochondrial elimination and mitophagy [10]. Thus, its precise function remains unclear.

To date, *ENDOG* impairment has been associated with an elevated left ventricular mass (LVM) related to the risk of heart failure and cardiac dysfunction, based on knock-out animal models and transcriptomic data from human hearts [11]. Nevertheless, despite *ENDOG* being suggested as a novel determinant of cardiac hypertrophy, no clearly pathogenic variants or genetically defined patients linked to this gene have been described. 

Here, we report biallelic novel *ENDOG* variants, identified by NGS in a patient with progressive external ophthalmoplegia (PEO), mitochondrial myopathy and multiple mtDNA deletions in the muscle.

## 2. Materials and Methods 

DNA from the patient’s muscle was screened by a customized panel of 300 nuclear genes previously associated with mitochondrial disorders or candidate genes for these conditions (list available upon request), and probes for the entire mtDNA (Agilent SureSelect, Agilent Technologies, Inc.,Santa Clara, CA, USA), sequenced with a MiSeq instrument (Illumina Inc., San Diego, CA, USA). Alignment, annotation and filtering were performed as previously described [12]; interesting variants have been confirmed by Sanger sequencing. For detection of mtDNA large-scale deletion, the protocol we recently developed [13] was applied. Whole exome sequencing (WES) was then performed (Illumina Exome Panel 45 Mb, Illumina Inc., San Diego, CA, USA), as previously described [12]. In silico prediction tools (PolyPhen-2 http://genetics.bwh.harvard.edu/pph2/ accessed on 28 February 2022 and SIFT, https://sift.bii.a-star.edu.sg/ accessed on 28 February 2022) were used to predict functional effects of genomic variants found. The crystal protein structure of ENDOG (AF-Q14249-F1) was downloaded from AlphaFold (https://alphafold.ebi.ac.uk/ accessed on 28 February 2022). The molecular structure was handled with PyMOL (http://www.pymol.org, version 2.5.2, accessed on 28 February 2022) to perform in silico mutagenesis for the p.Arg245Leu and p.Thr246Ile variants. Mitochondrial DNA content and integrity were assayed in muscle- or fibroblast-derived DNA by Southern blot, long-range polymerase chain reaction (PCR) and quantitative PCR, as described [14]. RNAseq was performed on RNA extracted from fibroblasts, processed following the TruSeq Stranded mRNA protocol (Illumina).

mtDNA depletion/repopulation experiments were carried out as previously described in [15]. The Nextera XT DNA Library Prep protocol (Illumina) was used for mtDNA sequencing on long-range PCR amplicons, and heteroplasmy level analysis of variants was performed as recently described [13].

For the screening of *ENDOG*, we selected patients with multiple mtDNA deletions or a clinical diagnosis of PEO without a single large-scale mtDNA deletion, present in the database of our laboratories (Unit of Medical Genetics and Neurogenetics, Fondazione IRCCS Istituto Neurologico “C. Besta” and Neuromuscular and Rare Diseases Unit, Fondazione IRCCS Ca’ Granda Ospedale Maggiore Policlinico). Informed consent for genetic analysis was available for all investigated subjects. 

Respiratory chain activities of complexes I to V, as well as citrate synthase (CS), were measured using standard spectrophotometric methods in supernatants of 800× *g* muscle homogenate, suspended in 10 mM phosphate buffer (pH 7.4) [16]. Western blot was performed on muscle and fibroblast samples, with the following antibodies: monoclonal ENDOG (B-2) (sc-365359 Santa Cruz, Dallas, TX, USA), total OXPHOS Human WB Antibody Cocktail (ab110411, Abcam, Cambridge, UK), monoclonal GAPDH (MAB374, Millipore, Burlington, MA, USA), monoclonal ACTB (A2066, Merck KGaA, Darmstadt, Germany).

## 3. Patient Clinical Report

The patient (Pt), now 64 years old, presented with a lifelong history of fatigue, muscle cramps and exercise intolerance. His family history was unremarkable. Neurological examination revealed external ophthalmoparesis, bilateral ptosis and weakness of facial muscles, rhinolalia and severe proximal > distal strength reduction conditioning myopathic gait disturbances and the need for double arm support to stand up from a chair. EMG confirmed upper and lower limb myopathic changes on both proximal and distal sites. Recently over follow-up, he experienced mild dysphagia that required diet adjustments, and sleep apnea, managed with nocturnal continuous positive airway pressure (CPAP). He also reported bowel disorder, previously diagnosed as irritable bowel syndrome. Cardiological assessment documented normal findings. Blood analysis showed hyperCKemia and increased lactic acid. He underwent a muscle biopsy for further investigations. Informed consent for biochemical and genetic studies was obtained from the proband in agreement with the Declaration of Helsinki.

## 4. Results

Spectrophotometric analysis of the mitochondrial respiratory chain complexes in muscle homogenate showed reduced complex I, III and IV activities, whereas complex II values were in the control range. Citrate synthase (CS), a common marker of mitochondrial content, was significantly increased (Appendix A).

Southern blot and long-range polymerase chain reaction (LR-PCR) analysis on the muscle revealed multiple mtDNA deletions. To assess mtDNA quantitative alterations, we also carried out qPCR on muscle DNA: we found a higher mtDNA content in the patient compared with controls, probably due to a compensatory effect, while the 7S/mtDNA ratio appeared only slightly increased (Figure 1). The presence of mtDNA deletions was confirmed by targeted NGS on mtDNA, together with the increased mtDNA/nDNA ratio (Appendix A).

The analysis of nuclear genes in the customized targeted gene panel led to the identification of three heterozygous variants in the *ENDOG* gene (NM_004435.2) (Figure 2). The single-nucleotide deletion c.61delG (rs1030852507) causes a frameshift predicted to insert a premature stop codon (p.Glu21Argfs*141); it appeared to be very rare in the gnomAD database (MAF value of 0.028%). It is classified as VUS according to the ACMG criteria (PM2) [17]. The two missense variants—c.734G > T, p.Arg245Leu (rs61737988) and c.737C > T, p.Thr246Ile (rs61737987)—affect two adjacent amino acids in the nuclease domain of the ENDOG protein; by looking at the NGS reads, we observed that they were in cis on the same allele. Prediction tools provided conflicting opinions. PolyPhen-2 predicted the p.Arg245Leu variant as “probably damaging” while predicting p.Thr246Ile as “benign”; instead, according to the SIFT algorithm, both variants are “deleterious”. Furthermore, in the ClinVar database, both are classified as “benign” possibly because of their frequency (MAF value of 0.2%), but no functional data were reported. Similarly, both are classified as benign according to the ACMG criteria (PM2, BS2, BP6), mainly because these variants were observed in a homozygous state in population databases more than expected for disease. The two nucleotide substitutions are not predicted to have any impact on splicing, to alter acceptor or donor site scores and to create new putative splice sites. Structural prediction for the mutant allele containing Leu245-Ile246 showed evident changes in steric hindrance, and loss of diverse hydrogen bonds of the affected amino acids with surrounding residues, compared to the wild-type amino acid sequence (Appendix A). Sanger sequencing confirmed the variants and segregation in three other members of the family (sister, son and daughter of the patient), which allowed us to determine their allelic distribution (Figure 2A). 

A single heterozygous variant in *C1QBP* (NM_001212.3) was detected: it is a nucleotide substitution, c.1A > G p.(Met1?), affecting the first amino acid of the protein. It is classified as VUS (PM2, PVS1). Biallelic mutations of *C1QBP* are associated with a spectrum of mitochondrial disorders, including PEO/myopathy [18]; nevertheless, the gene was fully covered in the analysis, without any evidence of a second variant. The *C1QBP* variant was absent in other available family members. Finally, the targeted gene panel revealed a single heterozygous variant in *MRPS34* (NM_023936.1), c.94C > T, p.Gln32*; it is a known SNP (rs763672163), classified as pathogenic (PVS1, PM2, PP5). However, mutations in this gene are associated with Leigh syndrome, with autosomal recessive inheritance. Because of the different phenotype and the presence of a single variant, this gene was not considered linked to the clinical presentation of the patient.

We performed WES to exclude a possible genetic cause due to a gene not present in our panel. By considering the information on phenotype, no further candidate gene was identified. Prioritization lead again to the *MRPS34* and *C1QBP* variants (Appendix A); the top candidate rare variants from WES are available online. Notably, WES analysis missed *ENDOG* as a possible cause, since the variant c.61delG was not identified, because it was located in a region not captured by the WES probes.

To examine the possible deleterious effect of the *ENDOG* variants, we performed Western blot on muscle biopsy and skin fibroblast samples from the patient and controls. In both patient’s specimens, we observed the virtual absence of the ENDOG protein, with a very low amount detectable only using long-time exposure (Figure 2C,D); the same finding was obtained by immunofluorescence, with a clear, mitochondrial-localized ENDOG signal in control fibroblasts and only background noise of the ENDOG signal in the patient’s fibroblasts (Appendix A). Therefore, we concluded that the identified *ENDOG* variants are pathogenic, including the two missense variants. Western blot analysis showed global perturbation of the muscle steady-state level of both mitochondrial- and nuclear-encoded OXPHOS proteins (Figure 3). C1QBP (also known as p32) resulted in being partly decreased, but not absent, as in another subject with biallelic *C1QBP* variants (Appendix A), suggesting this is not the primary genetic defect in our patient. RNAseq performed on fibroblast RNA showed an overlapping transcript (*c9orf114*) with *ENDOG* that makes it difficult to understand the differential allele expression. However, the two variants in cis appear as “heterozygous” (50% mutant and wt reads), indicating the biallelic expression of the transcript. Unfortunately, no reads were present in the region of the frameshift variant, confirming the difficulty of capturing this GC-rich region as happened during WES (Appendix A). Quantitative PCR showed a reduction of 50% in the expression levels of the mRNA of *ENDOG* in fibroblasts, and cDNA sequencing on the full-length *ENDOG* transcript confirmed the biallelic and balanced expression of both missense and frameshift variants (Appendix A). We did not observe any evidence of aberrant splicing after amplification of the whole *ENDOG* transcript; additionally, a dedicated analysis of the NGS coverage data did not reveal the presence of exon deletions. All these findings indicate that both deletion and missense variants probably affect protein stability, in addition to causing reduced transcript expression.

Since the ENDOG protein is expressed in fibroblasts (and was absent in the patient’s cells), we used this patient-specific cellular model to characterize ENDOG function. First, we performed Southern blot and long-range PCR analysis, but we did not observe the presence of multiple mtDNA deletions in the patient’s fibroblasts. This could be explained by the tissue-specific accumulation of mtDNA damage. The amount of mtDNA was normal, but to better examine whether the ENDOG protein is involved in mtDNA replication, we forced this process in cells by performing an mtDNA depletion/repopulation experiment. We tested whether the absence of ENDOG would lead to a delay in the mtDNA recovery after EtBr treatment, as observed in mutations of other genes involved in mtDNA replication such as *RNASEH1,* which is associated with slow or virtually no recovery in human fibroblasts [15]. In all cell lines (mutant and wild type), we observed a decrease in the mtDNA amount to <20% after EtBr depletion treatment, and then a recovery in the mtDNA amount. While a patient with mutations in *RNASEH1*, used as positive control, showed a clear replicative defect, no significant meaningful delay in the recovery of mtDNA was observed in the *ENDOG* mutant patient compared with the controls (Figure 4A). 

Another possible and interesting function suggested for ENDOG regards the repair of mtDNA damage by substitution of the improper bases introduced by replication error or by external sources (e.g., oxidative stress) [19,20]. To detect if there are any mtDNA alterations in fibroblasts, we sequenced the whole mtDNA by using NGS, which is also able to detect variants with low heteroplasmy levels. Interestingly, we found a higher amount of mtDNA point variants above the 5% heteroplasmy level in the patient’s fibroblasts compared to the age-matched controls (Figure 4B). All the homoplasmic changes observed in fibroblasts were present in the muscle, defining the patient’s haplogroup (H4a1a4b). Muscle DNA contained the heteroplasmic (≈50%) m.564G > A which was not present in the patient’s fibroblasts or lymphocytes. This variant is not present in public databases (Mitomap, GnomAD). It is located within the major H-strand promoter region (545–567) of the noncoding control region, and thus its role is hardly predictable. However, this substitution has been reported in single fibers from aging extraocular muscles [21].

Looking at the database of our laboratories, we found 98 patients with multiple mtDNA deletions or a clinical diagnosis of PEO who underwent NGS-based genetic analysis (targeted panel or WES), but none presented variants in the *ENDOG* gene. Furthermore, mutational analysis of *ENDOG* was performed in 65 patients with the same clinical and molecular features with no positive subject. Even the use of GeneMatcher (https://genematcher.org accessed on 28 February 2022) failed to identify additional cases.

## 5. Discussion

Mitochondrial DNA maintenance defects are a subgroup of mitochondrial diseases caused by an alteration in the nuclear genes involved in mtDNA maintenance. Usually, pathogenic variants in these genes lead to mtDNA depletion or multiple mtDNA deletions in affected organs due to defects in mtDNA synthesis or stability. These diseases follow a Mendelian inheritance (autosomal recessive or dominant) and are linked to various phenotypic presentations [22], including mitochondrial myopathies. PEO, myalgia, exercise intolerance and fatigue are frequent symptoms caused by primary mitochondrial myopathy (PMM). Accumulation of mtDNA deletions in the muscle is often a hallmark of these conditions, in particular those caused by mutations in genes encoding for proteins required for mtDNA replication and maintenance, such as *POLG, POLG2* and *TWNK* [22]. Notably, the same molecular and clinical features are associated with defects in *RNASEH1*, *MGME1* or *DNA2* [22], genes encoding mitochondrial nucleases, as well as *ENDOG*. The phenotype of the present patient with *ENDOG* variants overlaps classical PMM, with PEO as the main symptom and the presence of mtDNA deletions. It is important to note that in this patient, symptoms do not include heart involvement and impaired cardiac function, which have been considered the expected clinical output upon loss-of-function mutation in *ENDOG* [11]. An *ENDOG* null mouse has been reported [9]; it was viable and without any overt phenotype. Moreover, it showed no alteration in the mtDNA copy number, structure or mutation rate, but all the mitochondrial analyses were performed on the liver. A reduced amount of mtDNA in the heart was later described [11]. In contrast to the KO mouse model, our patient presents a clear myopathy phenotype and has multiple mtDNA deletions in the skeletal muscle that is strong evidence supporting a role in mtDNA metabolism for ENDOG. Moreover, these findings confirm that this tissue is the preferred site of accumulation of mtDNA deleted species, possibly due to mechanisms favoring the clonal expansion of mtDNA deletions [23], in particular in post-mitotic tissues [24]. A detailed analysis of the histological features in the muscle, together with a deeper investigation of the mtDNA, is recommended in the KO mouse model.

The absence of ENDOG in biological samples from the patient gave us the chance to characterize ENDOG function by comparing the patient’s fibroblasts with cells from healthy subjects or from patients who have known mtDNA replication impairment. We examined the possible involvement of ENDOG in mtDNA replication and concluded that its role in this process can be replaceable (under the conditions that we used in this study) or may be largely compensated by other factors or players, at least in fibroblasts. The possible involvement of ENDOG in mtDNA replication was previously investigated, providing evidence that ENDOG stimulates mtDNA replication initiation at O_H_ (origin of heavy strand replication) [5]. However, this stimulation was partially dependent on reactive oxygen species (ROS), suggesting a possible involvement of ENDOG in removal/repair of oxidatively damaged mtDNA nucleotides [5]. Given the close proximity of the mitochondrial genome to sites of ROS production, oxidative damage can occur at a higher frequency in mtDNA than in nuclear DNA. However, accumulation of defective mtDNA is prevented by the association of mtDNA with proteins (forming nucleoprotein structures called mitochondrial nucleoids), antioxidant molecules and base excision repair (BER) mechanisms. Along with BER enzymes, various endo/exonucleases may also function within mitochondria to selectively degrade damaged mtDNA copies with complex lesions [6,7]. It has been proposed that oxidative damage to mtDNA is not the major cause of somatic mtDNA mutations. The aging process contributes to the accumulation of many different point mutations with low heteroplasmy levels [25,26,27], but these variants occur mainly due to errors in DNA Pol-ɣ polymerase (POLG) during mtDNA replication [19,20]. Nevertheless, a study proposed a role for ENDOG in the removal of oxidatively damaged mtDNA when the BER machinery elements are impaired or the oxidative damage in mtDNA is severe [5]. Accordingly, our results indicate an accumulation of somatic point mutations in the patient’s fibroblasts, which advocates a role for ENDOG in the substitution/repair of the uncorrected bases. The different mechanisms for repairing point mutations in mtDNA are not completely known [28]; most of the studies report enzymes or pathways well described for nuclear DNA repair, also present inside mitochondria. Moreover, it is still debated how the mitochondrion degrades damaged mtDNA. It has been reported that ENDOG has an important role in mtDNA depletion, since it promotes cleavage of mtDNA as a response to oxidative stress, with a subsequent upregulation of mtDNA replication [5]. Even for mtDNA deletion formation, both replication and repair seem to play a role since nucleotide repeats, secondary structures and also mtDNA damage are associated with this genetic alteration [29]. In addition to the direct effect on mtDNA replication and repair, several different mitochondrial processes can be affected by excessive DNA damage and/or inefficient DNA repair, including mitochondrial dynamics and mitophagy [28]. All these different pathways need to be further and more deeply investigated to fully understand the role(s) of ENDOG in mtDNA metabolism. Despite the virtual absence of the ENDOG protein, the patient presented a late-onset disease, affecting only the muscle. In agreement with the absence of any overt phenotype in the mouse *ENDOG* null model, our data suggest that ENDOG is largely dispensable in most of the cell types and for most of life. As for other genetic forms of mitochondrial myopathies, the accumulation of multiple mtDNA deletions is mainly restricted to the skeletal muscle and associated with late-onset symptoms. Notably, the same molecular defect is also observed in normal aging post-mitotic tissues, with a preferred localization in extraocular muscles [21]. It is possible to speculate that the absence of ENDOG together with aging leads mtDNA deleted species to reach the threshold for disease appearance only in advanced age. Another hypothesis to explain the patient’s phenotype and the extreme rarity of this genetic condition is linked to the presence of a single deleterious variant in *C1QBP*. The exact function of C1QBP has not been defined, but the protein has a mitochondrial localization and diverse putative roles in mitochondrial function. Above all, biallelic mutations in *C1QBP* are associated with a spectrum of mitochondrial disorders, including myopathy and PEO. Despite not being enough to cause the disease, the presence of reduced C1QBP could have facilitated the clinical manifestation associated with the lack of ENDOG.

## 6. Conclusions

In conclusion, we identified biallelic variants in *ENDOG*, a gene that has not been previously associated with any mitochondrial disease. Additional patients are required to confirm the association between *ENDOG* pathogenic variants and mitochondrial myopathy and multiple mtDNA deletions; however, the identification of a subject with the virtual absence of ENDOG casts doubt on the assertion that loss-of-function mutations in *ENDOG* are associated with impaired cardiac function. Although the functional results obtained are preliminary, we provide novel evidence about a possible role of ENDOG linked to mtDNA maintenance. More studies are needed to further test involvement of ENDOG in mtDNA metabolism, not limited to replication but also to the complex repair systems for mtDNA, which are still poorly understood.

## Figures and Tables

**Figure 1 cells-11-00974-f001:**
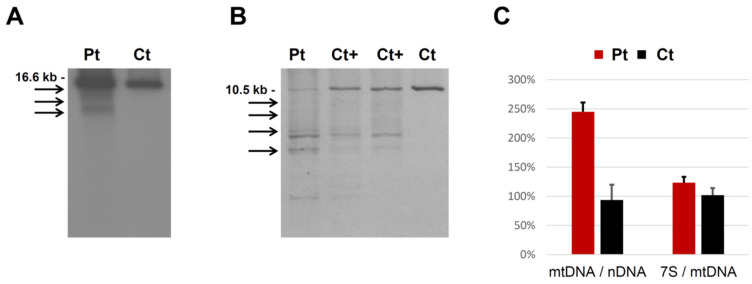
mtDNA studies in the patient (Pt) and controls (Cts). (**A**) Southern blot analysis of mtDNA from muscle samples. Arrows indicate multiple mtDNA deleted species. (**B**) Long-range PCR analysis of mtDNA (wild-type amplicon: 10.5 kbp). Arrows indicate multiple mtDNA deleted species; “Ct+” indicates samples obtained from patients with biallelic *POLG1* and *C1QBP* mutations, used as positive controls. (**C**) Quantitative PCR analysis of muscle mtDNA content normalized to nuclear DNA (mtDNA/nDNA) and 7S mtDNA levels normalized to total mtDNA in patient (Pt, red) and control (Ct, black) samples. Error bars indicate standard deviation.

**Figure 2 cells-11-00974-f002:**
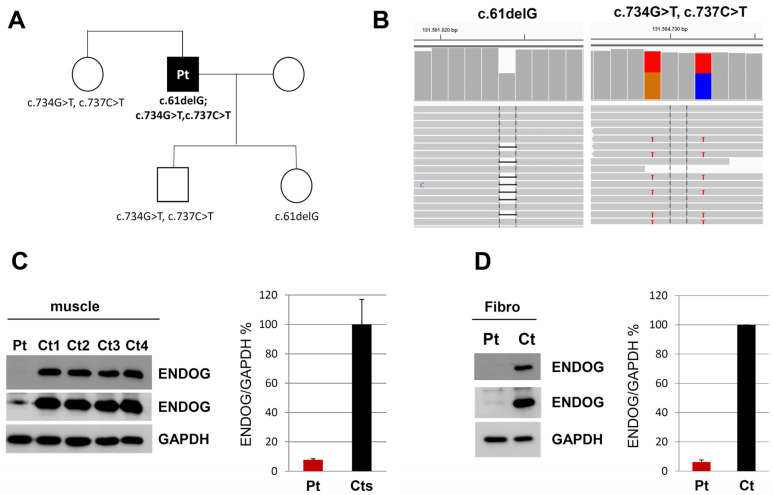
Genetic and protein studies on ENDOG. (**A**) Pedigree of the patient’s family, with segregation analysis of the *ENDOG* variants. (**B**) Snapshots from IGV software (Version 2.3.72) of the *ENDOG* variants identified in the proband. The changes c.734G > T and c.737C > T were always present on the same reads, indicating they are on the same allele. (**C**) Western blot analysis of ENDOG in skeletal muscle lysates from control individuals (Ct) and the patient described in the manuscript (Pt). Two different exposure times were used for the ENDOG antibody (shorter exposure in the upper panel, longer exposure in the middle panel). GAPDH (lower panel) was used as a loading control, and for normalization of the ENDOG amount. Error bars indicate standard deviation. (**D**) Western blot analysis of ENDOG in fibroblasts from a control individual (Ct) and the patient (Pt). Two different exposure times were used for the ENDOG antibody (shorter exposure in the upper panel, longer exposure in the middle panel). GAPDH (lower panel) was used as a loading control, and for normalization of the ENDOG amount. Error bars indicate standard deviation.

**Figure 3 cells-11-00974-f003:**
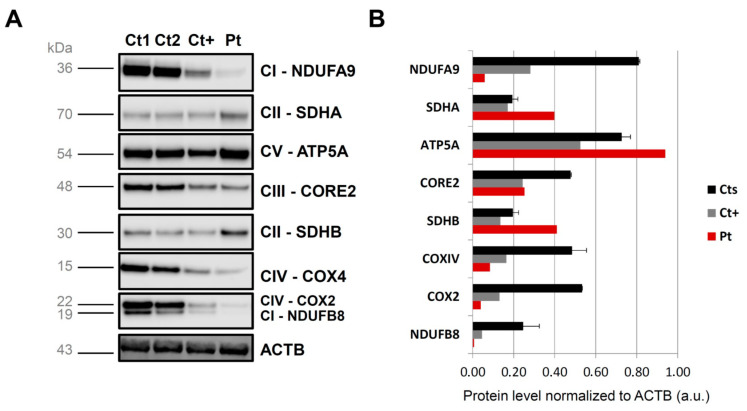
Steady-state levels of OXPHOS complex subunits. (**A**) Western blot analysis of OXPHOS subunits in skeletal muscle lysates from control individuals (Cts), a patient harboring biallelic *C1QBP* mutations (Ct+) and the patient described in the manuscript (Pt). OXPHOS subunit-specific antibodies were used against NDUFB8 or NDUFA9 (CI); SDHA or SDHB (CII); UQCRC2 (CIII); COXII or COXIV (CIV); and ATP5A (CV). Cytosolic b-actin (ACTB) was used as a loading control. (**B**) Graph reporting protein level normalized to ACTB (in percentage, 100% being the mean value of controls), for the Western blot depicted in panel A.

**Figure 4 cells-11-00974-f004:**
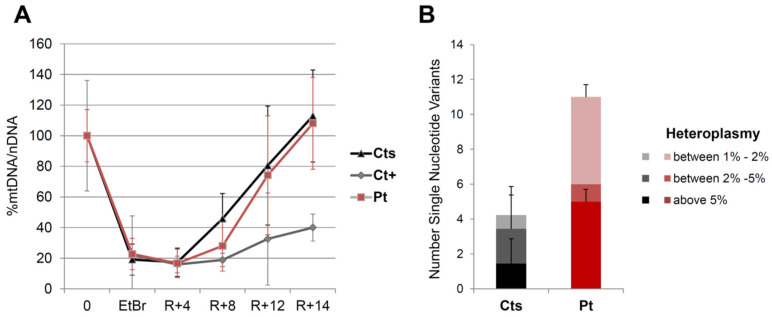
Functional studies on the patient’s fibroblasts. (**A**) Depletion/repopulation experiment in fibroblasts from the patient (Pt), a control individual (Ct) and a patient harboring biallelic *RNASEH1* mutations (Ct+). Depletion was achieved with the addition of EtBr to the culture medium for 4 days (EtBr), and recovery was followed up for 14 days (R + 14) after the removal of the drug. The percentage of mtDNA/nDNA was determined by qPCR; for each subject, an initial value was set up to 100%, before starting treatment (0). (**B**) Number of heteroplasmic variants detected by NGS in whole mtDNA from the patient (Pt) and control individuals (Ct). Three different ranges of heteroplasmy were considered: above 5%, between 2 and 5%, between 1 and 2%.

## Data Availability

Data supporting the reported results (vcf file of the targeted NGS; csv of the top 50 rare variants from WES; csv of WES rare variants prioritized by phenotype) can be found online here: https://doi.org/10.5281/zenodo.6033815, accessed on 28 February 2022.

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
