# Peer review of "Biallelic Variants in ENDOG Associated with Mitochondrial Myopathy and Multiple mtDNA Deletions"

_cells, 2022, doi:10.3390/cells11060974_

Round 1
Reviewer 1 Report
The authors have conducted a good and comprehensive work on describing the long term effects of ENDOG deficiency, this study is relevant in the global understanding of mitochondrial physiology. Even if the clinical impact of the study may not be quite relevant, their work does open the door to novel questions and further research. I truly believe that the authors should hypothesise a bit more and suggest further ways of research on this matter, maybe in animal models, but specially in being able to gather, compare and analyse international datasets that include clinical features, both retrospectiva and with a follow up that, in an ideal situation, could go on to the anatomy pathological analysis of post mortem samples, when the time comes.
On the experimental level, could you provide the CS quantification data? did you normalise it for number of cells? for protein level?
Reviewer 2 Report
In this study, the authors report biallelic novel ENDOG variants, identified by NGS in a patient with progressive external ophthalmoplegia (PEO), mitochondrial myopathy and multiple mtDNA deletions on muscle. Although these variants were predicted to be VUS, the authors detected a virtual absence of ENDOG protein in both the patient’s muscle biopsy and skin fibroblasts, which strongly suggests the pathogenicity. Although the patient’s fibroblasts did not present multiple mtDNA deletions or delay in replication process, an accumulation of low-level heteroplasmy mtDNA point mutations was observed compared with age-matched controls, suggesting a possible role of ENDOG in mtDNA replication or repair. There are a few minor points the authors may consider:
- The methods part could be re-organized to separate DNA from protein studies, each experiment should be briefly introduced with necessary details. Spectrophotometric analysis of the Mitochondrial Respiratory Chain complexes in muscle homogenate should also be included.
- The pathogenicity of these two in cis missense variants are still uncertain, given their relatively high allele frequency in population database. Did the authors check if these variants may cause splicing defects? Or are there any signs of exon deletion from panel and exome sequencing?
- Besides the presence of mtDNA deletions, did the authors check if there are any likely pathogenic mtDNA variants in the patient’s skeletal muscle?
